# Genetic characteristics of Jiaji Duck by whole genome re-sequencing

**Lihong Gu[1], Feng Wang[1], Zhemin Lin[1], Tieshan Xu[2]\*, Dajie Lin[1], Manping Xing[1], Shaoxiong Yang[1], Zhe Chao[1], Baoguo Ye[1], Peng Lin[3], Chunhui Hui[1], Lizhi Lu[4], Shuisheng Hou[5]**

**1** Institute of Animal Science & Veterinary, Hainan Academy of Agricultural Sciences, Haikou, P. R. China, **2** Tropical Crops Genetic Resources Institute, Chinese Academy of Tropical Agricultural Sciences, Haikou, P. R. China, **3** Hainan Chuanwei Muscovy Duck Breeding Co., Ltd, Wenchang, P. R. China, **4** Institute of Animal Science & Veterinary, Zhejiang Academy of Agricultural Sciences, Hangzhou, P. R. China, **5** Institute of Animal Sciences, Chinese Academy of Agricultural Sciences, Beijing, P. R. China

\* xutieshan760412@163.com

**Data Availability Statement:** Duck genomic resequencing datasets from JJ, FF, YD, BD and HD have been uploaded to the Short Read Archive (SRA) under the accession number PRJNA549423.

## Abstract

Jiaji Duck (JJ) is a Muscovy duck species that possesses many superior characteristics, and it has become an important genetic resource in China. However, to date, its genetic characteristics and genetic relationship with other duck breeds have not been explored yet, which greatly limits the utilization of JJ. In the present study, we investigated the genome sequences of 15 individual ducks representing five different duck populations, including JJ, French Muscovy duck (FF), mallard (YD), hong duck (HD) and Beijing duck (BD). Moreover, we investigated the characteristics of JJ-specific single nucleotide polymorphisms (SNPs) and compared the genome sequences of JJ vs. YD and JJ vs. BD using integrated strategies, including mutation detection, selective screening, and Gene Ontology (GO) analysis. More than 40 Gb of clean data were obtained for each population (mean coverage of 13.46 Gb per individual). A total number of 22,481,367 SNPs and 4,156,829 small insertion-deletions (Indels) were identified for the five duck populations, which could be used as molecular markers in breeding and utilization of JJ. Moreover, we identified 1,447,932 JJ-specific SNPs, and found that genes covering at least one JJ-specific SNP mainly involved in protein phosphorylation and dephosphorylation, as well as DNA modification. Phylogenetic tree and principal components analysis (PCA) revealed that the genetic relationship of JJ was closest to FF, while it was farthest to BD. A total of 120 and 111 genes were identified as positive selection genes for JJ vs. BD and JJ vs. YD, respectively. GO and Kyoto Encyclopedia of Genes and Genomes (KEGG) analyses showed that the positive selection genes for JJ vs. BD ducks mainly involved in pigmentation, muscle contraction and stretch, gland secretion, and immunology, while the positive selection genes obtained from JJ vs. YD ducks mainly involved in embryo development, muscle contraction and stretch, and gland secretion. Taken together, our findings enabled us to better understand the characteristics of JJ and provided a molecular basis for the breeding and hybrid utilization of JJ in the future.

**Funding:** The commercial company of Hainan Chuanwei Muscovy breeding Ltd is a commercial breeding company in Jiaji Duck. In this study, this company provided the blood samples of JJ Duck. This company did not provide support in the form of salaries for any author and did not have any additional role in the study design, data collection and analysis, decision to publish, or preparation of the manuscript.

**Competing interests:** There is no competing interest exist among authors and among affiliations those involved in this study. Hainan Chuanwei Muscovy breeding Ltd provided the blood samples of JJ Duck. This does not alter our adherence to PLOS ONE policies on sharing data and materials.

## 1. Introduction

Jiaji Duck (JJ), a Muscovy duck variety, is initially raised at Jiaji Town, Qionghai County, Hainan Province. JJ duck is covered by black and white plumage and has been believed to be originated from Southeast Asia. JJ has been raised at Jiaji Town for more than 200 years, and it has become an important genetic resource in China. JJ duck has various superior characteristics, such as fast growth, heat resistance, strong adaptability, high lean meat rate, and good meat quality. However, only few studies on JJ have been reported, while none of them is related to the genetic characteristics of JJ and genetic relationship between JJ and other duck breeds. This greatly limits the breeding and hybrid utilization of JJ. The approach of whole genome resequencing (WGRS) has been proved to be a very powerful tool in genetic evaluation and genetic relationship exploration of various animals, including pigs [1], chickens [2], cattle [3], sheep [4], dogs [5] and rabbits [6]. For ducks, the genome sequences have been constructed in 2013 [7], providing a platform for the research of ducks using WGRS method and promoting the related exploration of ducks. Xu et al. have explored the genetic signatures of artificial selection using genome WGRS data [8]. Zhang et al. have indicated the genomic variation in Peking duck populations using WGRS data [9]. Zhou et al. have identified the genes associated with body size and plumage color in ducks using WGRS and genome-wide association study (GWAS) technologies [10]. However, no report has investigated the genetic characteristics of JJ and its relationship with other duck populations using WGRS method.

In this study, we selected five duck populations that are different in a couple of characteristics. The first one was JJ, which has been introduced as above-mentioned. Similar to JJ, French Muscovy duck (FF) is another Muscovy duck population that has been introduced into Hainan Province since 1990's. However, FF grows faster than JJ, and it is fully covered by white plumage. Mallard (YD), widely accepted as the ancestor of domestic duck [11], is used to confirm whether JJ and FF are also derived from YD. Beijing duck (BD) has been subjected to intensive selection to provide the raw material for Beijing roast duck dishes. Hong duck (HD) is the hybrid offspring of Beijing duck and Sheldrake, which has higher quality of meat relative to Beijing duck and faster growth rate compared with Sheldrake. Four out of the five duck populations (JJ, FF, HD and BD) have been recorded by *Animal Genetic Resources in China Poultry* [12]. To characterize the genetic background of JJ duck and explore its genetic relationship with the above-listed duck breeds, we selected 15 ducks representing the five duck populations with three individuals for each population, and the whole genome of each duck was individually sequenced. Moreover, we, for the first time, investigated the genetic characteristics of JJ through analyzing JJ-specific single nucleotide polymorphisms (SNPs) and compared the genomic sequences of these five duck populations. Collectively, our results provided basis for the reasonable utilization of JJ in the future.

## 2. Materials and methods

### 2.1. Sample preparation

In the present study, 15 adult female ducks representing five different duck populations were selected, with three ducks for each population. Genomic DNA was extracted from about 5 mL venous blood for each individual as previously described [13]. After blood collection, the birds were housed for further analysis. JJ and FF ducks, obtained from Hainan Chuanwei Muscovy Duck Breeding Co., Ltd., were sampled. BD and HD ducks were developed by Institute of Animal Science, Chinese Academy of Agricultural Sciences. YD was raised by the Ji'ao Austrian Agricultural Science and Technology Co., Ltd. in Fenghua, Zhejiang Province.

**Table 1. The re-sequencing and mapping results.**

| Items | Clean Reads (M) | Clean Bases (Gb) | Mapped Bases (Gb) | Mapping Rate (%) | [1]Dup. Rate (%) | [2]Uniq. Rate (%) | Mean Depth | [3]Cov. Rate (%) |
|---|---|---|---|---|---|---|---|---|
| BD1 | 88.67 | 13.30 | 12.51 | 94.04 | 25.38 | 94.37 | 12.09 | 96.51 |
| BD2 | 90.83 | 13.63 | 12.79 | 93.86 | 24.93 | 93.88 | 12.39 | 96.85 |
| BD3 | 91.35 | 13.70 | 12.85 | 93.80 | 25.04 | 93.86 | 12.45 | 96.85 |
| FF1 | 90.35 | 13.55 | 12.45 | 91.84 | 23.53 | 93.96 | 12.32 | 94.10 |
| FF2 | 91.14 | 13.67 | 12.48 | 91.29 | 23.54 | 93.65 | 12.43 | 94.01 |
| FF3 | 90.41 | 13.56 | 12.48 | 92.06 | 23.99 | 93.93 | 12.33 | 94.09 |
| HD1 | 90.78 | 13.62 | 12.82 | 94.13 | 7.81 | 94.41 | 12.38 | 97.00 |
| HD2 | 90.07 | 13.51 | 12.91 | 95.52 | 24.99 | 95.23 | 12.28 | 96.51 |
| HD3 | 84.46 | 12.67 | 12.01 | 94.76 | 24.29 | 94.52 | 11.52 | 96.80 |
| JJ1 | 90.21 | 13.53 | 12.41 | 91.71 | 24.40 | 93.60 | 12.30 | 94.11 |
| JJ2 | 90.26 | 13.54 | 12.35 | 91.24 | 23.06 | 93.61 | 12.31 | 94.02 |
| JJ3 | 89.62 | 13.44 | 12.18 | 90.61 | 23.70 | 92.97 | 12.22 | 94.13 |
| YD1 | 89.14 | 13.37 | 12.57 | 93.98 | 25.15 | 94.42 | 12.15 | 96.50 |
| YD2 | 88.75 | 13.31 | 12.51 | 93.94 | 25.49 | 94.29 | 12.10 | 96.46 |
| YD3 | 89.99 | 13.50 | 12.72 | 94.25 | 24.95 | 94.51 | 12.27 | 96.50 |
| Means | 89.74 | 13.46 | 12.54 | 93.14 | 23.35 | 94.08 | 12.24 | 95.63 |
| Total | 1346.03 | 201.90 | 188.03 | | | | 195.78 | |

[1] Duplication rate

[2] Unique rate

[3] The coverage rate, referring to the percentage of the mapped bases divide by the total bases of duck genome, the mapped bases means bases that covered by cleans bases more than 1 X. BD1- BD3, Beijing duck, individual 1-individual 3; FF1- FF3, French Muscovy duck, individual 1- individual 3; HD1- HD3, Hong duck, individual 1-individual 3; JJ1- JJ3, Jiaji Duck, individual 1-individual 3; YD1- YD3, mallard, individual 1- individual 3.

## 2.2. Library construction and sequencing

Libraries were generated for each individual using standard Illumina sequencing protocols. The constructed libraries were sequenced as 150-bp paired-end reads on Illumina sequencing platform (HiSeqTM 4000), and more than 13 Gb clean data were generated for each individual (Table 1).

## 2.3. Read mapping

Sequencing adaptors were first removed using Trim Galore version 0.3.7 (http://www. bioinformatics.babraham.ac.uk/projects/trim_galore/). Subsequently, the trimmed data (clean data) were mapped to the duck genome assembly (ensemble version: BGI_duck_1.0) using the Burrows-Wheeler Alignment tool (BWA) [14] with default settings. The uniquely mapped reads were used for downstream analysis.

## 2.4. Detection and annotation of SNPs and small insertion–deletions (Indels)

For each individual of a breed, the genome analysis toolkit (GATK) was used to detect SNPs and Indels (1–50 bp) with GATK best practices method using RealignerTargetCreator, Indel-Realigner, HaplotypeCaller and GenotypeGVCFs [15]. Then SelectVariants was used to separate SNPs, Indels and other variants [16, 17]. The SNPs and Indels from the three individuals were merged to form the SNP and Indel sets of this breed. The merged SNPs and Indels were filtered using the parameters recommended by the GATK mentor, and those variants with

ultra-high ($>$ 500) or ultra-low ($<$ 4) coverage were discarded. The remaining variants were used for downstream analysis.

SnpEff [18] was used to annotate SNPs and Indels against the reference genome annotation to identify the effects of sequence variants on gene functionality.

## 2.5. The genetic characteristics of JJ obtained through analyzing JJ-specific SNPs and the related genes

If an SNP was only found in JJ population, it was considered as JJ-specific SNP without considering whether it was a homozygous or heterozygous SNP. To enhance the sufficiency, a JJ-specific SNP must be found in two of the three individuals of JJ ducks. According to these criteria, the JJ-specific SNPs were identified using a homemade pearl script. The SNPs that were located on the exons of duck genes were then selected and then classified into different types based on the annotation of SNPs, such as synonymous mutation SNPs, nonsynonymous mutation SNPs, stop-gain SNPs and stop-loss SNPs. Finally, the genes covering at least one JJ-specific SNP that was located on exons of genes were picked out, and Gene Ontology (GO) and Kyoto Encyclopedia of Genes and Genomes (KEGG) functional enrichment analyses were performed as described in section 2.8.

## 2.6. Genetic structure analysis of five duck populations

The genetic relationships of the five duck populations were inferred through phylogenetic tree analysis and principal components analysis (PCA). SNPs for each duck population with a minor allele frequency (MAF) $\leq$ 0.1 were removed, and the remaining SNPs were used for phylogenetic tree analysis and PCA. For phylogenetic tree analysis, NJ method of PHYLIP software was used to construct phylogenetic tree [19] with the default settings, and Newick Utilities software was used to generate phylogenetic tree graphics [20]. For PCA, EIGENSOFT software [21] was used to infer genetic relationships among the five duck populations, and the first two principal components were used to construct the PCA figure.

## 2.7. Analysis of selective sweeps, positive selection genes

Selective sweeps occur when the frequency of beneficial genetic variants is increased as a result of positive selection along with nearby linked genetic variants [9]. Generally, positive selection can lead to reduced heterozygosity (Hp) of the selected population and increased fixation index (Fst) between populations around the selected site [6]. In order to define candidate regions undergoing positive selection, outlier approach in combination with Hp estimates was used for the five sequenced duck populations with estimates of Fst between duck populations. Since the current duck reference genome was assembled in sub-chromosomal scaffolds [22], only scaffolds$>$150 kb (1,226 scaffolds) were used for the downstream analysis to ensure the accuracy of our calculation of Hp and Fst, which was similar to studies on rabbits [6] and chickens [2]. In addition, to reduce the number of false positive windows, windows with $>$ 20 SNPs were used in downstream analysis. The Hp values and Fst values of the genomic regions were calculated using a 100 kb-sliding window (step = 50 kb), Z-transform Hp (ZHp) and FST (ZFst) as previously described [5]. Windows with both their ZHp and ZFst values falling in the top 5% were identified as the positive selection regions. Genes that were overlapped with or harbored by positive selection regions were defined as positive selection genes and were used in downstream analysis.

## 2.8. GO and KEGG analyses of positive selection genes

The duck GO annotation information from the Ensembl Genome Browser was used to perform GO enrichment analysis of genes under positive selection, and the whole genome set of protein-coding genes in the annotation was used as the background. For each query, represented GO terms were tested against the background using the GOstats Bioconductor R package (https://www.r-project.org/) [23]. To further understand the biological functions of the positive selection genes, KEGG [24] was used to perform pathway enrichment analysis.

## 2.10. Funding

This work was financially supported by Key Research and Development Programs of Hainan Province (Grant no. ZDYF2019053, ZDYF2018224), Chinese Modern Technology System of Agricultural Industry (Grant no. CARS-42-50), and Central Public-interest Scientific Institution Basal Research Fund for Chinese Academy of Tropical Agricultural Sciences (Grant no. 1630032017034).

## 2.11. Ethics statement

The animal-related handling and sampling procedures were approved by the Animal Care and Use Committee of Chinese Academy of Tropical Agricultural Sciences (CATAS), and great efforts were made to minimize the suffering of animals in accordance with recommendations of European Commission (1997). All methods were conducted in accordance with relevant guidelines. Venous blood was kindly collected to ameliorate the suffering of ducks.

## 3. Results and discussion

### 3.1. A number of variants are found in duck genome

WGRS data were obtained from five duck populations with three samples of each population (JJ1, JJ2, JJ3; BD1, BD2, BD3; FF1, FF2, FF3; HD1, HD2, HD3; YD1, YD2, YD3) using HiSeq TM 4000 system. After quality control [25], averagely 13.46 Gb (12.24 X) data were obtained for each individual, > 40 Gb (> 36 X) data were obtained for each duck population, and a total of 201.90 Gb (195.78 X) high-quality data remained (Table 1). Subsequently, the high-quality data were mapped to the *A. platyrhynchos* genome (BGI_duck_1.0), resulting in an average mapping rate of 93.14% and an average coverage rate of 95.63% for each individual (Table 1). The average mapping rate of this study (93.14%) was higher compared with our previous report (89.71%) [8], indicating a better quality of sequencing data used in this study.

In this study, SNPs and Indels of each duck population were identified using GATK according to previous studies [26, 27]. The SNPs and Indels for codons with multiple alleles were removed. Since the duck genome only contains scaffolds without chromosomes, we could not compare the genome differences among the five duck populations at the chromosomal level. For JJ duck, a total number of 22,481,367 SNPs and 4,156,829 Indels were identified after filtration. For SNPs, ~50% of total SNPs were located in intergenic regions (11,042,847 SNPs, 49.12%), and 45.31% of total SNPs were located in intronic regions (10,186,307). SNPs located in exonic regions (303,498) only accounted for 1.35% of the total SNPs (Fig 1 and Table 2). Similar to SNPs, 50.02% of total Indels (2,079,246) were located in intergenic regions, 44.02% of total Indels (1,874,314) were located in intronic regions, and only 1.44% of total Indels (59,858) were harbored by exonic regions (Fig 1 and Table 2). These results indicated that the majority of the identified mutations were located in non-coding sequences (intergenic regions or intronic regions), and only a small proportion of them fell in exon sequence, showing that only very few variants along with duck genome could exert functional effects on protein

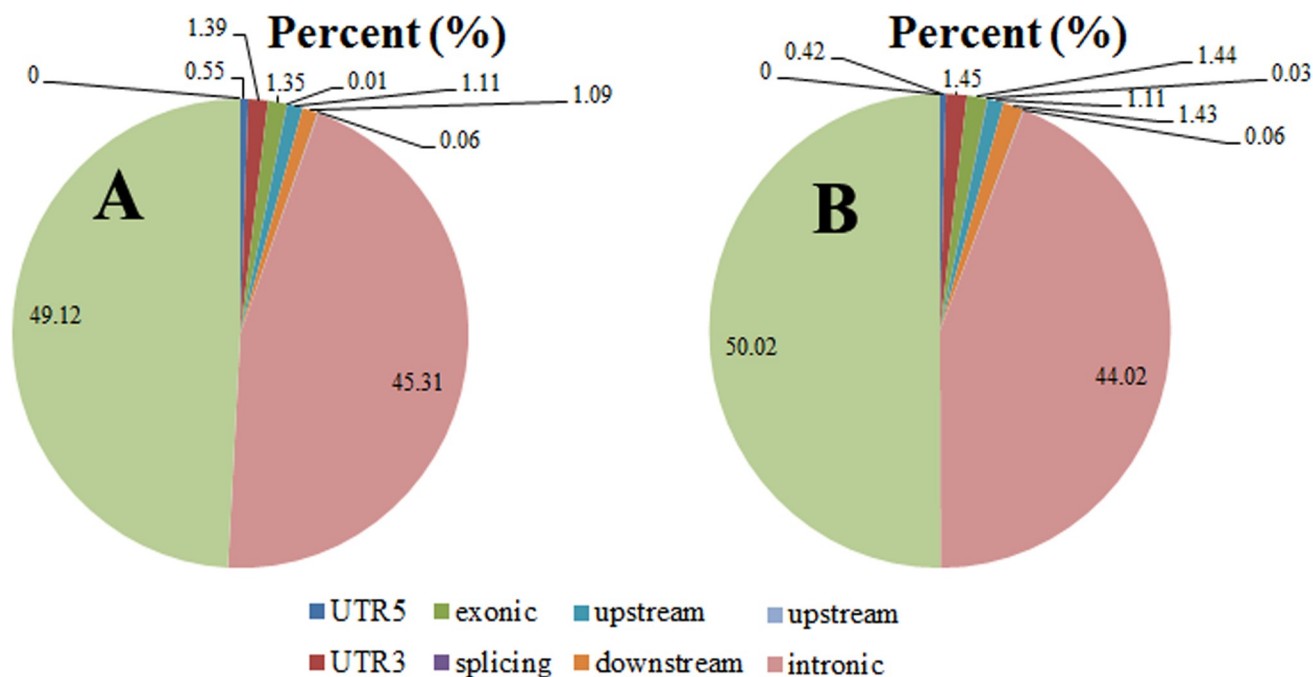

**Fig 1. The percentage of SNPs and Indels of JJ located in different regions of duck genome.** (A) The percentage of SNPs which fell in different regions of duck genome; (B) The percentage of Indels which fell in different regions of duck genome.

function. Our results were consistent with those of Xu et al. [8] in ducks and Carneiro et al. in rabbits [6].

**Table 2. Annotation of identified SNPs and Indels for JJ duck population.**

| Items | SNPs | | Indels | |
|---|---|---|---|---|
| | Number | Percent (%) | Number | Percent (%) |
| Total | 22,481,367 | 100 | 4,156,829 | 100 |
| UTR5 | 123,648 | 0.55 | 17,459 | 0.42 |
| UTR3 | 312,491 | 1.39 | 60,274 | 1.45 |
| exonic | 303,498 | 1.35 | 59,858 | 1.44 |
| splicing | 2,248 | 0.01 | 1,247 | 0.03 |
| exonic; splicing | 0 | 0 | 0 | 0 |
| upstream | 249,543 | 1.11 | 46,141 | 1.11 |
| downstream | 245,047 | 1.09 | 59,443 | 1.43 |
| upstream | 13,489 | 0.06 | 2,494 | 0.06 |
| intronic | 10,186,307 | 45.31 | 1,829,836 | 44.02 |
| intergenic | 11,042,847 | 49.12 | 2,079,246 | 50.02 |
| ncRNA_UTR3 | 0 | 0 | 0 | 0 |
| ncRNA_UTR5 | 0 | 0 | 0 | 0 |
| ncRNA_exonic | 0 | 0 | 0 | 0 |
| ncRNA_splicing | 0 | 0 | 0 | 0 |
| ncRNA_intronic | 2,248 | 0.01 | 831 | 0.02 |
| other | 0 | 0 | 0 | 0 |

SNPs mean single nucleotide polymorphisms; Indels mean short insertions and deletions.

## 3.2. The JJ-specific SNPs

The specific SNPs are usually the basic genetic characteristics of a species. A high throughput SNP marker can offer strong accuracy for genotyping, which can be used in marker-assisted selection [28, 29]. In this study, we screened the JJ-specific SNPs and found 1,447,932 specific SNPs, of which only 56,651 SNPs (accounting for 4% of the total specific SNPs) were located in the exons of duck genes. Among these SNPs located in the exons, there were 36,517 synonymous mutation SNPs (64.46%), suggesting that the majority of the SNPs were random mutations that did not lead to the changes of amino acids. In addition, there were 19,817 nonsynonymous mutation SNPs, which could lead to the changes of amino acids. Moreover, we detected 285 stop-gain SNPs and 11 stop-loss SNPs, indicating that SNP mutations resulting in alteration or loss of gene function were rare events, which was consistent with previous reports in rabbits [6], pigs [30], and chickens [2].

The genes covering at least one JJ-specific non-synonymous SNP could lead to functional differences of these genes for JJ compared with other duck populations, and herein might represent the specificity of JJ. In this study, we selected genes covering at least one JJ-specific nonsynonymous SNP, and performed GO and KEGG analysis on these genes to explore JJ-specific genetic characteristics. The biological process (BP) of GO analysis demonstrated that genes covering at least one JJ-specific non-synonymous SNP mainly played roles in protein phosphorylation and dephosphorylation, as well as DNA modification (Fig 2A and S1 Table). We also carried out KEGG analysis and constructed the significant enrichment graphics of q-values via the 20 most significant pathways (Fig 2B and S2 Table). The results indicated that Fanconi anemia pathway was the most significant pathway, while the metabolic pathway covered the most genes. These findings might reflect the specificity of JJ in Fanconi anemia and metabolism.

## 3.3. Genetic relationships between JJ and other duck populations

A phylogenetic tree is a branching diagram or "tree" showing the inferred evolutionary relationships among various biological species or subspecies based on the similarities and differences of their genetic characteristics [8]. In this study, we constructed the phylogenetic tree to illustrate the genetic relationships among the five duck populations (Fig 3A). The results showed that the individuals of JJ were firstly clustered into a class, followed by FF, YD, HD, and BD sequentially, indicating that FF was the closest population genetically related to JJ, followed by YD, HD and BD. Interestingly, the genetic relationship between JJ and YD was closer than that between YD and HD or BD. Since YD (mallard) was considered as the ancestor of HD and BD, we hypothesized that JJ and FF were differently originated from HD and BD. In addition, we used PCA to infer genetic relationships among the five duck populations in another aspect (Fig 3B). The PCA results revealed that individuals of JJ and FF were clustered into one group, suggesting the closest genetic relationship of JJ and FF, which was consistent with the results of phylogenetic tree analysis. Individuals in the remaining three populations were clustered into three separate groups.

## 3.4. Many genomic regions and genes associated with positive selective between JJ and other duck populations are found

Selective sweeps occur when the frequency of beneficial genetic variants is increased as a result of positive selection along with nearby linked genetic variants [9]. Given that the current duck reference genome is assembled in sub-chromosomal scaffolds [22], only scaffolds>150 kb (1,226 scaffolds) were used for the downstream analysis in our study to ensure the accuracy of

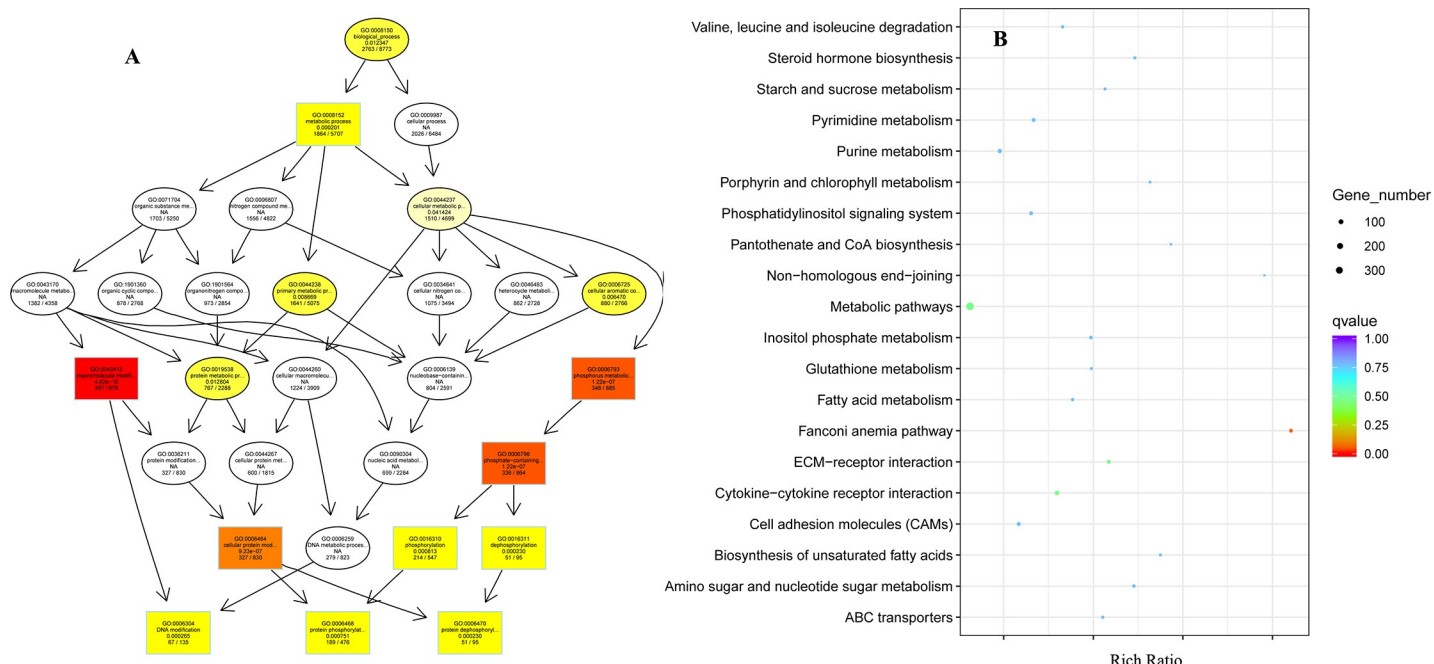

**Fig 2. The functional enrichment graphics for genes covering at least one JJ-specific non-synonymous SNP. (A)** Directed acyclic graph of GO enrichment analysis. Colors from light to dark represent significant levels from weak to strong, respectively. **(B)** The significant enrichment graphics of q-values using the 20 most significant pathways.

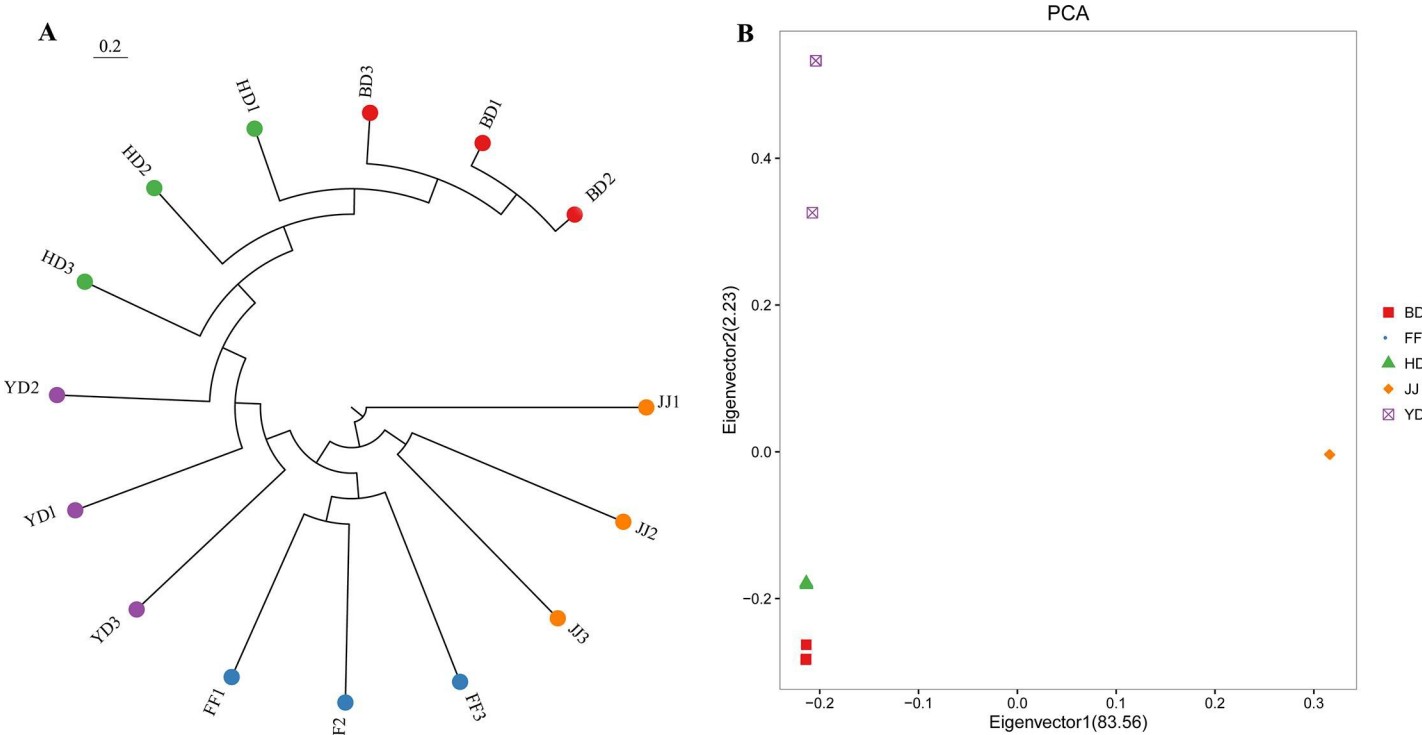

**Fig 3. Population genetic structure of five duck populations. (A)** Results of phylogenetic tree analysis. A kind of color represents a duck population, and a big dot represents a duck individual. **(B)** PCA plot of duck populations. Eigenvector 1 and 2 explained 83.56% and 2.23% of the observed variance, respectively.

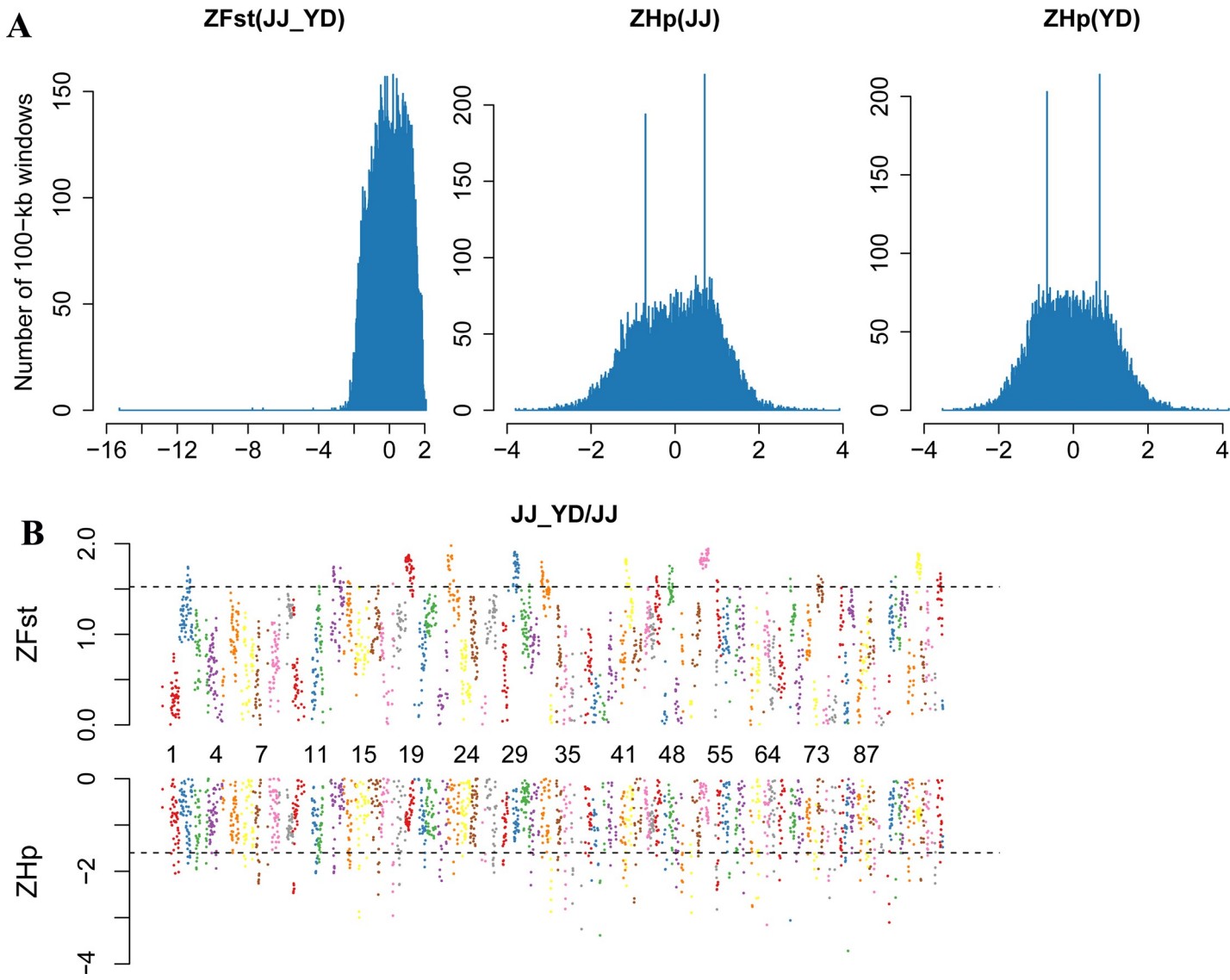

**Fig 4. Selective sweep analysis of the duck genome of JJ-YD.** (**A**), Distribution of ZFst in JJ-YD, ZHp in JJ, and ZHp in YD for all 100-kb windows. Bins of ZFst and ZHp are presented along the x axes. (**B**), The positive end of the ZFst(JJ-YD) distribution and the negative end of the ZHp(JJ) distribution.

Hp and Fst calculation, which was similar to studies on rabbits [6] and chickens [2]. We calculated the Hp and Fst values of the genomic regions using a 100 kb-sliding window (step = 50 kb). Windows with both their ZHp and ZFst values in the top 5% were identified as the positive selection regions. Since individuals in JJ and FF were clustered into one group, and genetic structure analysis suggested that HD was very close to BD in genetics, we only explored the positive selection regions in two comparisons, namely JJ vs. YD and JJ vs. BD. We selected the longest 100 scaffolds to construct the ZHp and ZFst figures to show the basic information. For JJ vs. YD comparison, 79 positive selection regions were obtained, which covered 111 genes (Fig 4 and S3 Table). For JJ vs. BD comparison, 76 positive selection regions were obtained, which covered 120 genes (Fig 5 and S4 Table).

The GO project is a collaborative effort to develop and use ontology to support biologically meaningful annotation of genes and their products [31]. In this study, we performed GO

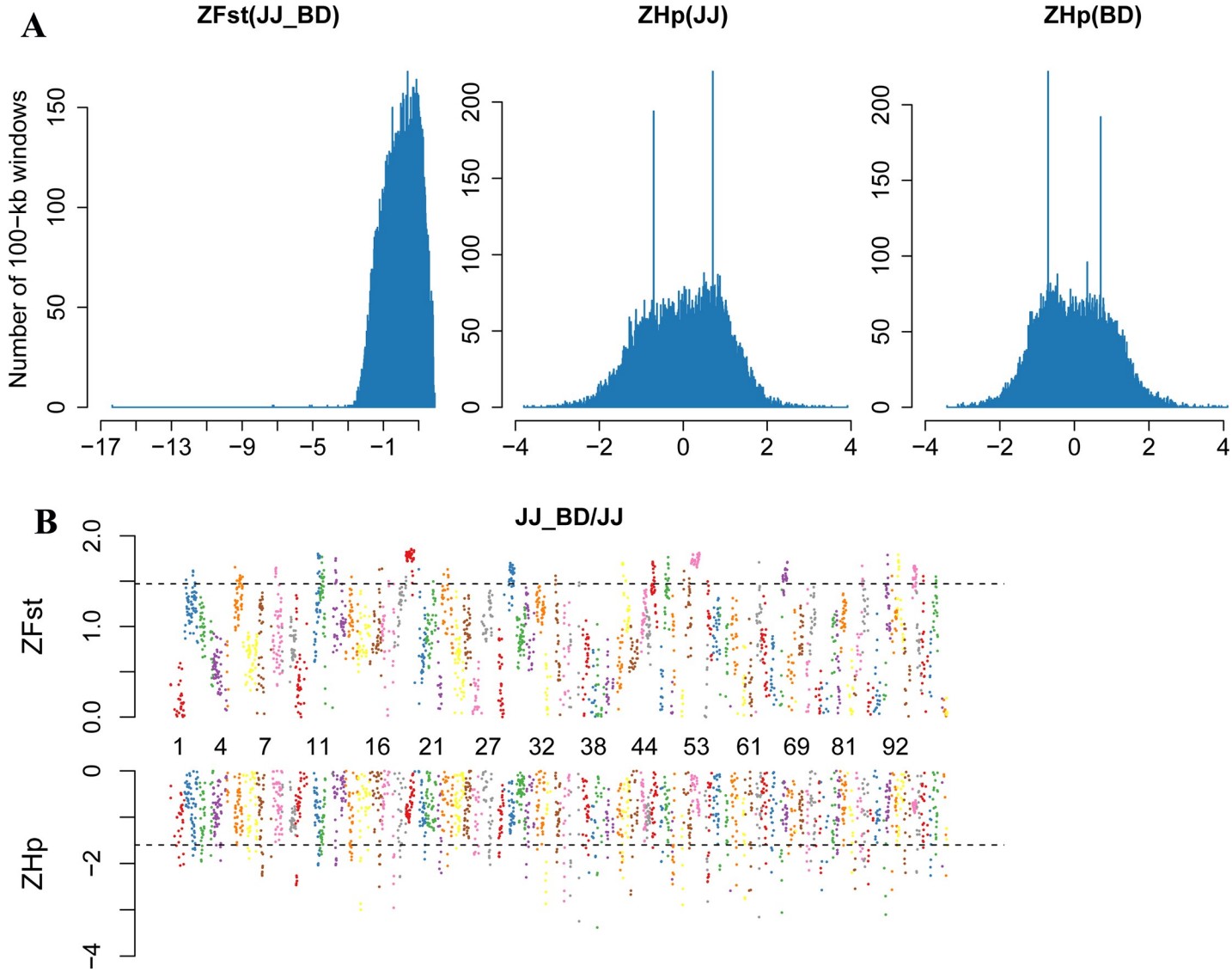

**Fig 5. Selective sweep analysis of the duck genome of JJ-BD.** (**A**), Distribution of ZFst(JJ-BD), ZHp(JJ) and ZHp(BD) for all 100-kb windows. Bins of ZFst and ZHp are presented along the x axes. (**B**), The positive end of the ZFst(JJ-BD) distribution and the negative end of the ZHp(JJ) distribution.

analysis to describe the properties of positive selection genes of JJ vs. BD ducks and those of JJ vs. YD ducks. For positive selection genes of JJ vs. BD ducks, 232 significant GO terms were enriched (S5 Table). Of which, the 20 most significant terms (Fig 5) were mainly involved in pigmentation (terms included melanocyte differentiation, pigment cell differentiation, developmental pigmentation), muscle contraction and stretch (muscle contraction, muscle system process, detection of muscle stretch, Wnt signaling pathway [32, 33], skeletal muscle thin filament assembly), and immunology (negative regulation of mast cell apoptotic process, mast cell apoptotic process, regulation of mast cell apoptotic process, mast cell homeostasis) [34, 35]. The results were consistent with the differences in feather color, athletic ability, and immunity between JJ and YD. For positive selection genes of JJ vs. YD ducks, 152 significant GO terms were obtained (S6 Table). The positive selection genes of JJ vs. YD ducks were mainly involved in embryo development because eight GO terms were related to embryo development in the

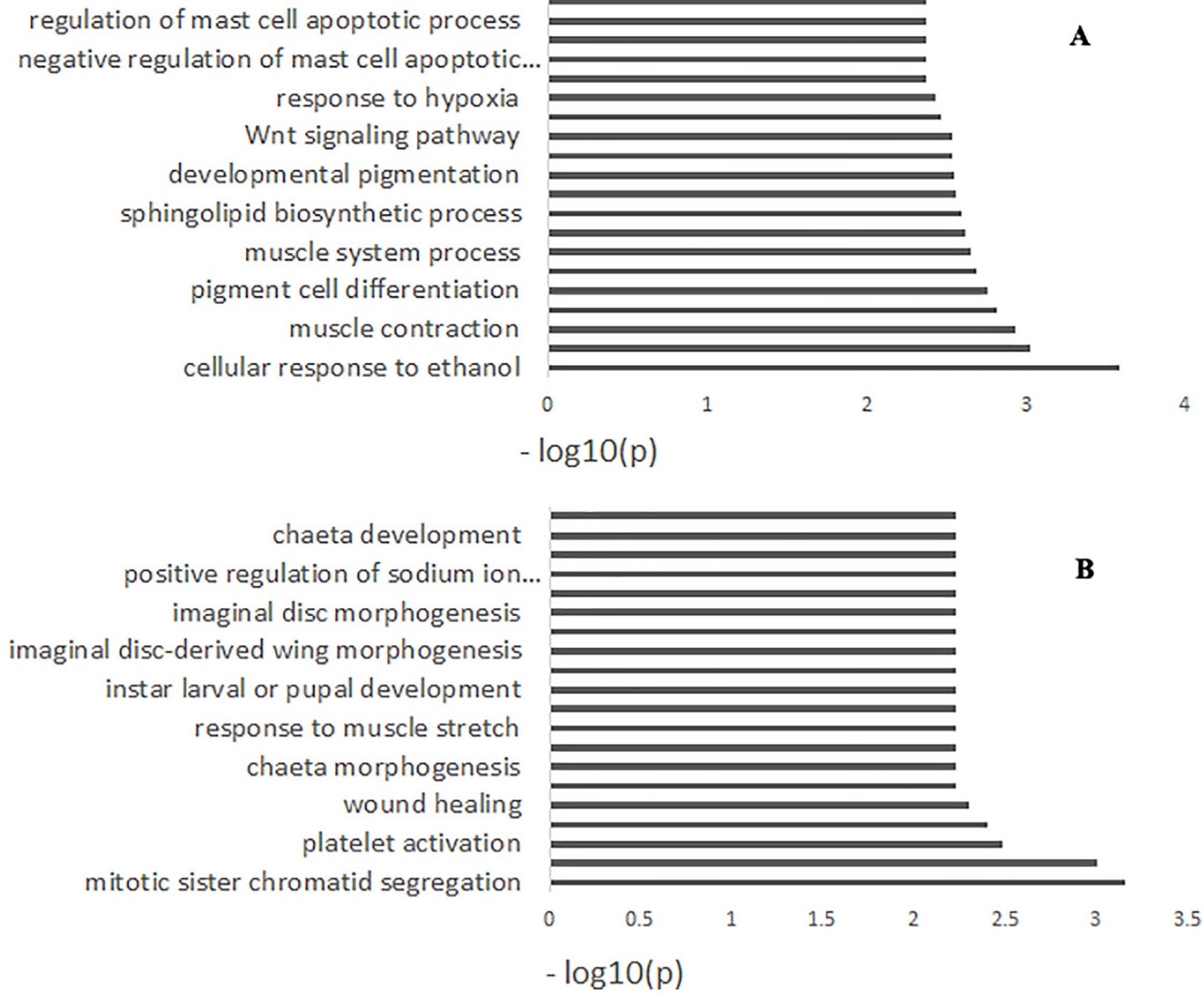

**Fig 6. The 20 most significant annotation pathways of genes located in the top 5% ZFst and ZHp. (A)** for JJ vs. BD ducks, **(B)** for JJ vs. YD ducks. The x-axis shows the -log10(P-value) of each pathway.

20 most significant terms, including wing disc development, instar larval or pupal development, larval development, imaginal disc-derived wing morphogenesis, wing disc morphogenesis, imaginal disc morphogenesis, post-embryonic appendage morphogenesis, and instar larval or pupal morphogenesis. Subsequently, terms of muscle contraction and stretch (muscle contraction, response to muscle stretch, and detection of muscle stretch) were also enriched (Fig 6). These results highlighted the differences in embryo development and muscle contraction and stretch between JJ and YD.

KEGG is a "computer representation" of the biological system [24], and the KEGG database can be utilized for modeling, simulation, browsing and retrieval of data. In this study, the significant pathways (P<0.05) for JJ vs. BD ducks and JJ vs. YD ducks were selected (Table 3).

**Table 3. Significantly enriched KEGG pathways for JJ vs. BD ducks and JJ vs. YD ducks.**

| JJ vs. BD | | |
|---|---|---|
| pathway names | p-value | genes |
| Hypertrophic cardiomyopathy (HCM) | 0.002826 | gene4589, gene16244, gene13399 |
| Salivary secretion | 0.003638 | gene13401, gene6509, gene12554 |
| Insulin secretion | 0.005433 | gene14642, gene6509, gene12554 |
| Pancreatic secretion | 0.006374 | gene13401, gene6509, gene12554 |
| Bile secretion | 0.03918 | gene14642, gene12554 |
| Dilated cardiomyopathy | 0.0404 | gene4589, gene13399 |
| Gastric acid secretion | 0.0404 | gene13401, gene12554 |
| PI3K-Akt signaling pathway | 0.042055 | gene3589, gene9998, gene16244 |
| **JJ vs. YD** | | |
| Salivary secretion | 0.002882 | gene13401, gene6509, gene12554 |
| Insulin secretion | 0.004314 | gene14642, gene6509, gene12554 |
| Pancreatic secretion | 0.005068 | gene13401, gene6509, gene12554 |
| Aldosterone-regulated sodium reabsorption | 0.010621 | gene9288, gene12554 |
| Jak-STAT signaling pathway | 0.015062 | gene9287, gene3589, gene9286 |
| Ubiquitin mediated proteolysis | 0.024208 | gene12556, gene12904, gene9290 |
| Hypertrophic cardiomyopathy (HCM) | 0.028686 | gene4589 gene13399 |
| Bile secretion | 0.03379 | gene14642, gene12554 |
| Cytokine-cytokine receptor interaction | 0.034 | gene9287, gene3589, gene9286 |
| Dilated cardiomyopathy | 0.034851 | gene4589, gene13399 |
| Gastric acid secretion | 0.034851 | gene13401, gene12554 |

Grey represents the comparison of JJ vs. BD and the secretion related pathways in the significant enriched KEGG pathways. Light green means the comparison of JJ vs. BD and the secretion related pathways in the significant enriched KEGG pathways.

The results showed that the pathways related to gland secretion were significantly enriched for JJ vs. BD and JJ vs. YD. For example, five out of eight significantly enriched pathways were associated with gland secretion for JJ vs. BD, while five out of 11 significantly enriched pathways for JJ vs. YD ducks were related with gland secretion. The results of KEGG analysis showed that gland secretion was the main difference between JJ and BD as well as YD.

Collectively, we, for the first time, explored the genetic characteristics of JJ and its genetic relationship with other four duck breeds using WGRS data. The findings of this study would benefit our understanding of the genetic characteristics of JJ and provide valuable references for the utilization of JJ breeds in the future.

# Supporting information

**S1 Table. The BP of GO analysis for genes covering at least one JJ-specific non-synonymous SNP.**
(XLSX)

**S2 Table. The KEGG analysis for genes covering at least one JJ-specific non-synonymous SNP.**
(XLSX)

**S3 Table. Positive selection genomic regions and genes for JJ vs. YD ducks.**
(XLSX)

**S4 Table. Positive selection genomic regions and genes for JJ vs. BD ducks.**
(XLSX)

**S5 Table. Significantly enriched GO terms for positive genes of JJ vs. BD ducks.**
(XLSX)

**S6 Table. Significantly enriched GO terms for positive genes of JJ vs. YD ducks.**
(XLSX)

## Acknowledgments

We thank Dr. Li Chen for providing mallard blood and egg samples, Prof. Qi Zhang and Dr. Zhanbao Guo for their help with sample collection, Dr. Kyle Schachtschneider for editing the manuscript and Dr. Minggang Wang for modifying the initial manuscript.

## Author Contributions

**Conceptualization:** Tieshan Xu, Shuisheng Hou.

**Data curation:** Zhemin Lin.

**Formal analysis:** Zhe Chao.

**Funding acquisition:** Lihong Gu.

**Investigation:** Feng Wang, Dajie Lin.

**Methodology:** Baoguo Ye.

**Project administration:** Manping Xing.

**Software:** Lizhi Lu.

**Supervision:** Shaoxiong Yang.

**Validation:** Chunhui Hui.

**Visualization:** Peng Lin.

**Writing – original draft:** Lihong Gu.

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
