## [Decision Letter · Decision Letter 0]

15 Oct 2019

PONE-D-19-18411

Genetic characteristics of Jiaji Duck as revealed by genome resequencing data

PLOS ONE

Dear Mr Xu,

Thank you for submitting your manuscript to PLOS ONE. After careful consideration, we feel that it has merit but does not fully meet PLOS ONE’s publication criteria as it currently stands. Therefore, we invite you to submit a revised version of the manuscript that addresses the points raised during the review process.

We would appreciate receiving your revised manuscript by Nov 29 2019 11:59PM. To enhance the reproducibility of your results, we recommend that if applicable you deposit your laboratory protocols in protocols.io, where a protocol can be assigned its own identifier (DOI) such that it can be cited independently in the future. For instructions see: http://journals.plos.org/plosone/s/submission-guidelines#loc-laboratory-protocols

We look forward to receiving your revised manuscript.

Kind regards,

Bi-Song Yue, Ph.D

Academic Editor

PLOS ONE

Journal Requirements:

2) In your Methods, please state the volume of the blood samples collected for use in your study.

3) In your Methods section, please include a comment about the state of the animals following this research. Were they euthanized or housed for use in further research? If any animals were sacrificed by the authors, please include the method of euthanasia and describe any efforts that were undertaken to reduce animal suffering.

4) We note that you are reporting an analysis of a microarray, next-generation sequencing, or deep sequencing data set. PLOS requires that authors comply with field-specific standards for preparation, recording, and deposition of data in repositories appropriate to their field. Please upload these data to a stable, public repository (such as ArrayExpress, Gene Expression Omnibus (GEO), DNA Data Bank of Japan (DDBJ), NCBI GenBank, NCBI Sequence Read Archive, or EMBL Nucleotide Sequence Database (ENA)). In your revised cover letter, please provide the relevant accession numbers that may be used to access these data. For a full list of recommended repositories, see http://journals.plos.org/plosone/s/data-availability#loc-omics or http://journals.plos.org/plosone/s/data-availability#loc-sequencing.

5) Thank you for stating the following in the Competing Interests section:

[The authors have declared that no competing interests exist.].   

We note that one or more of the authors are employed by a commercial company: Hainan Chuanwei Muscovy breeding Ltd

i. Please provide an amended Funding Statement declaring this commercial affiliation, as well as a statement regarding the Role of Funders in your study. If the funding organization did not play a role in the study design, data collection and analysis, decision to publish, or preparation of the manuscript and only provided financial support in the form of authors' salaries and/or research materials, please review your statements relating to the author contributions, and ensure you have specifically and accurately indicated the role(s) that these authors had in your study. You can update author roles in the Author Contributions section of the online submission form.

ii. Please also provide an updated Competing Interests Statement declaring this commercial affiliation along with any other relevant declarations relating to employment, consultancy, patents, products in development, or marketed products, etc. 

Reviewers' comments:

Reviewer's Responses to Questions

**Comments to the Author**

1. Is the manuscript technically sound, and do the data support the conclusions?

Reviewer #1: Partly

Reviewer #2: Yes

Reviewer #3: Yes

2. Has the statistical analysis been performed appropriately and rigorously? 

Reviewer #1: Yes

Reviewer #2: Yes

Reviewer #3: Yes

3. Have the authors made all data underlying the findings in their manuscript fully available?

Reviewer #1: Yes

Reviewer #2: Yes

Reviewer #3: Yes

4. Is the manuscript presented in an intelligible fashion and written in standard English?

Reviewer #1: No

Reviewer #2: Yes

Reviewer #3: Yes

5. Review Comments to the Author

Reviewer #1: This paper involves the sequencing of several individuals (3 per breed) spanning 5 duck breeds for the purpose of identify SNPs specific to the Jaiji duck (JJ) and to then broadly characterize the genes that are under positive selection.

The basis of the paper is sound and overall the science seems reasonable with the caveats below.

However there are two major issues that need to be corrected:

1) Numerous issues with grammar that range from trivial to humorous to obscuring the meaning of the sentence. I would recommend edited specifically for correct use of English.

2) The description of how the SNPs were called and how JJ specific snps were identified is incomplete. For example, it is not clear if SNPs are called on each individual or on the pooled reads per breed (or something else). This makes understanding the methodology difficult because the parameter space is not very clear. The authors describe that JJ specific SNPs were identified using a homemade perl script but neither the approach nor the criteria used in the script were described at all. Combined, these issues would make replicating the study impossible. It also leaves to the imagination what a JJ specific SNP would be. Are they homozygous and found in all three individuals or is it sufficient if they are in one individual? What about SNPs on the sex chromosomes? This should be associated with a discussion of what categories of SNPs would or would not be identified given this approach. I also feel that this homemade script should be provided.

Some other more more minor points:

1) line 50 and 63 : please explicitly define utilization in these contexts. Sure information can be useful but used to do what? There is no reason for mystery here but the vagueness of the statement is quite noticeable.

2) Figure 1 does not really show anything nor does it indicate which breeds are in panel a or b. Table 2 should probably show SNP counts/frequencies for every breed. I presume that table 2 only shows the JJ breed data.

3) Figure 2b - label the x-axis

4) Figure 3a - there is no metric on the tree; Figure 3ab - these really don't show much - a Venn diagram (or equivalent) showing the number of SNPs per breed would be a much better way of demonstrating this information

5) Figure 4 & 5 and associated text - what portion of the genome is covered by the sweep analysis? Its not clear why a 5% cutoff was set for sweeps - this seems to imply that there WERE selective sweeps but what if there wasn't a sweep? there should probably be a supplemental table that explicitly defines these regions.

Reviewer #2: In this study, the authors selected five duck populations with three individuals of each population to sequence the whole genome of each duck. Using these resequencing data, the authors explored out 1,447,932 JJ-specific SNPs which can be used as molecular markers, performed the genetic characteristics research of JJ and the genetic relationship between JJ and other duck breeds. Overall, the results of this paper are very interesting and it is an excellent paper. However, there are still some minor problems that should be modified.

1. The results of genetic relationship among these five duck populations should be mentioned in the abstract;

2. The tables presented in this paper should be changed to three-line tables;

3. please list the age and sex for the sequencing duck;

4. The description of sequencer is inconsistent in this paper;

5. The language should be improved by a native English speaker in order to eliminate grammatical and spelling errors and to conform to correct scientific English. I listed some of these errors below.

Line 30-33: “However, to date, the genetic characteristics and genetic relationship with other duck breeds has not been explored yet,” is not clear. The authors should refer to whose “genetic characteristics and genetic relationship with other duck breeds”;

Line 91: “other” should be changed to “the”;

Line 102: A comma should be placed behind “JJ and FF ducks”;

Line 251-252: The authors should delete the “on these genes”;

Line 275: should add a comma between FF and YD;

Line 298:should add blank space between the number and KB.

Reviewer #3: In my view, the manuscript is meaningful for breeding and genetics in duck. But , there are several shortcomings.

L8： What is the ‘d’ means?

L55-63：Jiaji Duck, if it is a definite breed, I think you should provide the information of variety approval, like the record of Breeds of Domestic Animal and Poultry in China or other published researches. So should other breeds you used in your research.

L64-75：The introduction of technical methods, NGS or others, is not the point. You can review the research progress in duck sequencing.

Why you choose these four breeds (France muscovy ducks, mallard, hong duck, and beijing duck) and did a phylogenetic tree for the all five breeds?

Results and discussion：Only basic and necessary results were showed.

What puzzles me is this, you list your results and say they are meaningful, but I can not find any deep discussion. After your sequencing and analysis, some specific SNPs and Indel, GO and KEGG project can illustrate what? I think readers are look forward to in-depth discussions and the specific and functional point.

Tables：All your tables do not meet the standards (Three-line table) , required notes missing. In Table 3, what is the meaning of different background notes?

Reference ：Format is also nonstandard. Journal Title, abbreviation, PubMed Central PMCID and so on.

6. PLOS authors have the option to publish the peer review history of their article (what does this mean?). If published, this will include your full peer review and any attached files.

Reviewer #1: No

Reviewer #2: No

Reviewer #3: No

---

## [Author Response · Author response to Decision Letter 0]

14 Jan 2020

Journal Requirements:

A: We have changed the style of our manuscript to meet PLOS ONE's style requirements according to http://www.journals.plos.org/plosone/s/file?id=wjVg/PLOSOne_formatting_sample_main_body.pdf and http://www.journals.plos.org/plosone/s/file?id=ba62/PLOSOne_formatting_sample_title_authors_affiliations.pdf and marked the modifications in red in the revised manuscript. 

2. In your Methods, please state the volume of the blood samples collected for use in your study.

A: We have provided the volume of the blood samples collected for use in our study and marked in red in unmarked manuscript (LINE 94-95). That is “Genomic DNA was extracted from about 5 mL venous blood for each individual as previously described [13]”.

3. In your Methods section, please include a comment about the state of the animals following this research. Were they euthanized or housed for use in further research? If any animals were sacrificed by the authors, please include the method of euthanasia and describe any efforts that were undertaken to reduce animal suffering.

A: We inserted the sentence “After blood collection, the birds were housed for further analysis” in the revised manuscript (LINE 96) and marked in red.

4. We note that you are reporting an analysis of a microarray, next-generation sequencing, or deep sequencing data set. PLOS requires that authors comply with field-specific standards for preparation, recording, and deposition of data in repositories appropriate to their field. Please upload these data to a stable, public repository (such as ArrayExpress, Gene Expression Omnibus (GEO), DNA Data Bank of Japan (DDBJ), NCBI GenBank, NCBI Sequence Read Archive, or EMBL Nucleotide Sequence Database (ENA)). In your revised cover letter, please provide the relevant accession numbers that may be used to access these data. For a full list of recommended repositories, see http://journals.plos.org/plosone/s/data-availability#loc-omics or http://journals.plos.org/plosone/s/data-availability#loc-sequencing.

A: According to PLOS requirements, we comply with field-specific standards for preparation, recording, and deposition of data in repositories appropriate to their field. We then deposit the data obtained in this study to the Short Read Archive (SRA). The statements in revised manuscript were “2.9. Data availability

Duck genomic resequencing datasets from JJ, FF, YD, BD and HD were uploaded to the Short Read Archive (SRA) under the accession number of PRJNA549423” (LINE 174-177).

5) Thank you for stating the following in the Competing Interests section:

[The authors have declared that no competing interests exist.]. 

We note that one or more of the authors are employed by a commercial company: Hainan Chuanwei Muscovy breeding Ltd

i. Please provide an amended Funding Statement declaring this commercial affiliation, as well as a statement regarding the Role of Funders in your study. If the funding organization did not play a role in the study design, data collection and analysis, decision to publish, or preparation of the manuscript and only provided financial support in the form of authors' salaries and/or research materials, please review your statements relating to the author contributions, and ensure you have specifically and accurately indicated the role(s) that these authors had in your study. You can update author roles in the Author Contributions section of the online submission form.

ii. Please also provide an updated Competing Interests Statement declaring this commercial affiliation along with any other relevant declarations relating to employment, consultancy, patents, products in development, or marketed products, etc. 

A: We have provided the Funding Statement and Competing Interests Statement in our cover letter according to the journal requirements. 

Response to reviewers

Response to Reviewer #1

This paper involves the sequencing of several individuals (3 per breed) spanning 5 duck breeds for the purpose of identify SNPs specific to the Jaiji duck (JJ) and to then broadly characterize the genes that are under positive selection.

The basis of the paper is sound and overall the science seems reasonable with the caveats below.

However there are two major issues that need to be corrected:

1) Numerous issues with grammar that range from trivial to humorous to obscuring the meaning of the sentence. I would recommend edited specifically for correct use of English.

A: The revised manuscript was polished by Kyle Schachtschneider, a post doctor student from the University of Illinois in America. The additions and changes are marked in red in the revised text.

2) The description of how the SNPs were called and how JJ specific snps were identified is incomplete. For example, it is not clear if SNPs are called on each individual or on the pooled reads per breed (or something else). This makes understanding the methodology difficult because the parameter space is not very clear. The authors describe that JJ specific SNPs were identified using a homemade perl script but neither the approach nor the criteria used in the script were described at all. Combined, these issues would make replicating the study impossible. It also leaves to the imagination what a JJ specific SNP would be. Are they homozygous and found in all three individuals or is it sufficient if they are in one individual? What about SNPs on the sex chromosomes? This should be associated with a discussion of what categories of SNPs would or would not be identified given this approach. I also feel that this homemade script should be provided.

A: We appreciate the reviewer to point out the defects in calling the SNPs and identifying the JJ-specific SNPs. In the revised manuscript, we added the necessary information in calling the SNPs and identified the JJ-specific SNPs. The revised contents are listed as follow.

For SNPs calling, “2.4. Detection and annotation of SNPs and small insertion–deletions (Indels) 

For each individual of a breed, the genome analysis toolkit (GATK) was used to detect SNPs and Indels (1–50 bp) with GATK best practices method using RealignerTargetCreator, IndelRealigner, HaplotypeCaller and GenotypeGVCFs [15]. Then SelectVariants was used to separate SNPs, Indels and other variants [16, 17]. The SNPs and Indels from the three individuals were merged to form the SNP and Indel sets of this breed. The merged SNPs and Indels were filtered using the parameters recommended by the GATK mentor, and those variants with ultra-high (> 500) or ultra-low (< 4) coverage were discarded. The remaining variants were used for downstream analysis.” (LINE 112-121)

For the identification of JJ-specific SNPs, “2.5. The genetic characteristics of JJ obtained through analyzing JJ-specific SNPs and the related genes

If an SNP was only found in JJ population, it was was considered as JJ-specific SNP without considering whether it was a homozygous or heterozygous SNP. To enhance the sufficiency, a JJ-specific SNP must be found in two of the three individuals of JJ ducks. According to these criteria, the JJ-specific SNPs were identified using a homemade pearl script. The SNPs that were located on the exons of duck genes were then selected and then classified into different types according to the annotation of SNPs, such as synonymous mutation SNPs, nonsynonymous mutation SNPs, stop-gain SNPs and stop-loss SNPs. Finally, the genes covering at least one JJ-specific SNP that was located on exons of genes were picked out, and Gene Ontology (GO) and Kyoto Encyclopedia of Genes and Genomes (KEGG) functional enrichment analyses were performed as described in section 2.8 of this study.”. (LINE 124-136)

In the current paper, we performed our analysis based on duck genome assembly (ensemble version: BGI_duck_1.076). In this version of duck genome assembly, there are only scaffolds without chromosomes. Therefore, we did not explore the SNPs information on the sex chromosomes. We added the correlated discussion of what categories of SNPs would or would not be identified in our revised manuscript. We listed the related context. “Since the duck genome only contains scaffolds without chromosomes, we could not compare the genome differences among the five duck populations at the chromosomal level.” (LINE 213-215).

Some other more minor points:

1) line 50 and 63 : please explicitly define utilization in these contexts. Sure information can be useful but used to do what? There is no reason for mystery here but the vagueness of the statement is quite noticeable.

 A: This is a constructive suggestion. We changed “utilization” in line 50 and 63 of the initial manuscript to “breeding and hybrid utilization” in revised manuscript and marked in red (LINE 47 and 60).

2) Figure 1 does not really show anything nor does it indicate which breeds are in panel a or b. Table 2 should probably show SNP counts/frequencies for every breed. I presume that table 2 only shows the JJ breed data.

A: We think the suggestion is very suitable. Therefore, we replaced the data by the JJ breed data in Table 2. Correspondingly, the figure 1 was drawn using the SNPs and Indels frequencies of JJ duck in the revised manuscript. Figure 1A shows the distribution of SNPs along with duck genome, and Figure 1B shows the distribution of Indels. In addition, the related context that descripted Table 2 and Figure 1 was changed correspondingly. The changed items were listed as follow.

Table 2. Annotation of identified SNPs and Indels for JJ duck population

Items SNPs Indels

 Number Percent (%) Number Percent (%)

Total 22,481,367 100 4,156,829 100

UTR5 123,648 0.55 17,459 0.42

UTR3 312,491 1.39 60,274 1.45

exonic 303,498 1.35 59,858 1.44

splicing 2,248 0.01 1,247 0.03

exonic;splicing 0 0 0 0

upstream 249,543 1.11 46,141 1.11

downstream 245,047 1.09 59,443 1.43

upstream 13,489 0.06 2,494 0.06

intronic 10,186,307 45.31 1,829,836 44.02

intergenic 11,042,847 49.12 2,079,246 50.02

ncRNA_UTR3 0 0 0 0

ncRNA_UTR5 0 0 0 0

ncRNA_exonic 0 0 0 0

ncRNA_splicing 0 0 0 0

ncRNA_intronic 2,248 0.01 831 0.02

other 0 0 0 0

Note: SNPs mean single nucleotide polymorphisms; Indels mean short insertions and deletions.

The related context that descripted Table 2 and Figure 1. “In this study, SNPs and Indels of each duck population were identified using GATK according to previous studies [26, 27]. The SNPs and Indels for codons with multiple alleles were removed. Since the duck genome only contains scaffolds without chromosomes, we could not compare the genome differences among the five duck populations at the chromosomal level. For JJ duck, a total number of 22,481,367 SNPs and 4,156,829 Indels were identified after filtration. For SNPs, ~50% of total SNPs were located in intergenic regions (11,042,847 SNPs, 49.12%), and 45.31% of total SNPs were located in intronic regions (10,186,307). SNPs located in exonic regions (303,498) only accounted for 1.35% of the total SNPs (Fig. 1 and Table 2). Similar to SNPs, 50.02% of total Indels (2,079,246) were located in intergenic regions, 44.02% of total Indels (1,874,314) were located in intronic regions, and only 1.44% of total Indels (59,858) were harbored by exonic regions (Fig. 1 and Table 2). These results indicated that the majority of the identified mutations were located in non-coding sequences (intergenic regions or intronic regions), and only a small proportion of them fell in exon sequence, showing that only very few variants along with duck genome could exert functional effects on protein function. Our results were consistent with those of Xu et al. [8] in ducks and Carneiro et al. in rabbits [6].”

3) Figure 2b - label the x-axis

A: We labeled the x-axis for Figure 2b in the revised manuscript, which was listed as follow.

4) Figure 3a - there is no metric on the tree; Figure 3ab - these really don't show much - a Venn diagram (or equivalent) showing the number of SNPs per breed would be a much better way of demonstrating this information

A: We have added the metric on the tree (Fig 3A, we listed the revised Fig 3 at the end of question). Fig 3, including two panels (Fig 3A and Fig 3B), were used to illustrate the genetic relationships between JJ and other duck populations. Fig 3A, the results of phylogenetic tree analysis, indicated the evolutionary relationships of the five duck populations obviously. Fig 3B, the results of principal components analysis (PCA), also clearly showed the genetic relationships of these five duck populations. Therefore, we think Fig 3 can clearly illustrate the genetic relationships between JJ and other duck populations.

In our opinion, a Venn diagram can clearly show the SNPs number per breed which is really a better way to demonstrate specific SNP number and shared SNP number for the five duck populations. Thus, we demonstrated the genetic relationships between JJ and other duck populations using Fig 3 of this paper without using a Venn diagram.

5) Figure 4 & 5 and associated text - what portion of the genome is covered by the sweep analysis? Its not clear why a 5% cutoff was set for sweeps - this seems to imply that there WERE selective sweeps but what if there wasn't a sweep? there should probably be a supplemental table that explicitly defines these regions.

A: Previous studies have found that "selection on a single advantageous mutation affects patterns of genetic variation at nearby loci, causing (i) a reduction in heterozygosity, (ii) a skewed allele frequency distribution and (iii) an excess of high frequency derived alleles across a region surrounding the selected allele (Stephan, W. Philos T R Soc B 365, 1245-1253)". Hence a reduction in heterozygosity (Hp) and an increase in fixation index (Fst) were accordingly used as signatures to define regions underwent natural or artificial selection. This method has been well accepted and widely adopted. For example, in the study of the polygenic basis for phenotypic change during domestication of rabbit, the authors defined selection regions using an Fst-H outlier approach and set Fst > 0.35 and H< 0.05 as the criteria to define a selected region (Carneiro et al. Science, 2014, 345(6200):1074-1079). In addition, the authors of the study of dog’s adaptation to a starch rich diet defined selection regions using a ZFst (and/or ZHp) threshold of 5 (Axelsson et al. Nature, 2013, 495(7441):360-364.). In the current paper, we applied the Fst-H outlier approach with setting both ZHp values and ZFst values falling in the top 5% to identify the positive selection regions (those mean selective sweeps). Focusing on those regions, and by analyzing the function of their associated genes, we can get a clear idea on which aspects of biological functionality are among the most preferred targets of selection.

As a result, we provided the positive selection genomic regions and genes for JJ vs. YD ducks and for JJ vs. BD ducks as the supplemental Table 3 and supplemental Table 4.

References：

Stephan W. Genetic hitchhiking versus background selection: the controversy and its implications[J]. Philosophical Transactions of the Royal Society B Biological Sciences, 2010, 365(1544):1245-1253.

Carneiro M, Rubin CJ, Di Palma F et al. Rabbit genome analysis reveals a polygenic basis for phenotypic change during domestication[J]. Science, 2014, 345(6200):1074-1079.

Axelsson E, Ratnakumar A, Arendt M L, et al. The genomic signature of dog domestication reveals adaptation to a starch-rich diet [J]. Nature, 2013, 495(7441):360-364.

Response to Reviewer #2

In this study, the authors selected five duck populations with three individuals of each population to sequence the whole genome of each duck. Using these resequencing data, the authors explored out 1,447,932 JJ-specific SNPs which can be used as molecular markers, performed the genetic characteristics research of JJ and the genetic relationship between JJ and other duck breeds. Overall, the results of this paper are very interesting and it is an excellent paper. However, there are still some minor problems that should be modified.

1. The results of genetic relationship among these five duck populations should be mentioned in the abstract;

 A: We added the results of genetic relationship analysis in the abstract which is listed as “Phylogenetic tree and principal components analysis (PCA) revealed that the genetic relationship of JJ was closest to FF, while it was farthest to BD.” in the revised manuscript (LINE 37-39).

2. The tables presented in this paper should be changed to three-line tables;

 A: We changed all the tables in the paper to three-line tables.

3. please list the age and sex for the sequencing duck;

 A: We added “adult female” in “Sample preparation” in the revised manuscript (LINE 93).

4. The description of sequencer is inconsistent in this paper;

 A: We read through our paper again and found the sequencer was written as “HiSeqTM 4000” in line 109 of initial version, while “HiSeq 4000” in Line 199. Therefore, we changed the “HiSeq 4000” in Line 199 to “HiSeqTM 4000” in revised version(LINE 195).

5. The language should be improved by a native English speaker in order to eliminate grammatical and spelling errors and to conform to correct scientific English. I listed some of these errors below.

Line 30-33: “However, to date, the genetic characteristics and genetic relationship with other duck breeds has not been explored yet,” is not clear. The authors should refer to whose “genetic characteristics and genetic relationship with other duck breeds”;

 A: We have added “of JJ” behind the genetic characteristics and genetic relationship in the revised version. The modified sentence in revised manuscript is as follow: “However, to date, its genetic characteristics and genetic relationship with other duck breeds have not been explored yet, which greatly limits the utilization of JJ.” (LINE 22-24).

Line 91: “other” should be changed to “the”;

 A: We have changed “other” to “the” in the revised version and marked in red.

Line 102: A comma should be placed behind “JJ and FF ducks”;

 A: We have added a comma behind “JJ and FF ducks” in the revised manuscript.

Line 251-252: The authors should delete the “on these genes”;

 A: We have deleted the “on these genes”.

Line 275: should add a comma between FF and YD;

 A: We have added a comma between FF and YD.

Line 298:should add blank space between the number and KB.

 A: We have added a blank space between the number (100) and KB.

Response to Reviewer #3

In my view, the manuscript is meaningful for breeding and genetics in duck. But , there are several shortcomings.

L8： What is the ‘d’ means?

 A: The ‘d’ means nothing. We deleted the ‘d’ in the revised manuscript.

L55-63：Jiaji Duck, if it is a definite breed, I think you should provide the information of variety approval, like the record of Breeds of Domestic Animal and Poultry in China or other published researches. So should other breeds you used in your research.

A: This is a good suggestion. Mallard (YD) was widely accepted as the ancestor of domestic duck [11], which was indicated in the initial manuscript. The other four duck populations were all recorded by Breeds of Domestic Animal and Poultry in China. Therefore, we added a sentence as “Four out of the five duck populations (JJ, FF, HD and BD) have been recorded by Breeds of Domestic Animal and Poultry in China [12].” in the revised manuscript (LINE 81-83).

11. Li HF, Zhu WQ, Song WT, Shu JT, Han W, Chen KW. Origin and genetic diversity of Chinese domestic ducks. Mol Phylogenet Evol. 2010;57(2):634-640. doi: 10.1016/j.ympev.2010.07.011. PubMed PMID: 20674751.

12. Chinese animal and poultry genetic resources commission. Breeds of Domestic Animal and Poultry in China. 2010. 

L64-75：The introduction of technical methods, NGS or others, is not the point. You can review the research progress in duck sequencing.

 A: We agree with the opinion of the reviewer. Therefore, we deleted the NGS introduction and reviewed the research progress in duck sequencing which was listed as follow in revised version (LINE 60-71). “The approach of whole genome re-sequencing (WGRS) has been proved to be a very powerful tool in genetic evaluation and genetic relationship exploration of various animals, including pigs [1], chickens [2], cattle [3], sheep [4], dogs [5] and rabbits [6]. For ducks, the genome sequences have been constructed in 2013 [7], providing a platform for the research of ducks using WGRS method and promoting the related exploration of ducks. Xu et al. have explored the genetic signatures of artificial selection using genome WGRS data [8]. Zhang et al. have indicated the genomic variation in Peking duck populations using WGRS data [9]. Zhou et al. have identified the genes associated with body size and plumage color in ducks using WGRS and genome-wide association study (GWAS) technologies [10]. However, no report has investigated the genetic characteristics of JJ and its relationship with other duck populations using WGRS method.”.

Why you choose these four breeds (France muscovy ducks, mallard, hong duck, and beijing duck) and did a phylogenetic tree for the all five breeds?

 A: Actually, we chose five duck populations. We reread our paper again and found the vague statement about the duck populations. For example, we wrote that “In this study, we selected five duck populations that differ in a couple of characteristics. Similar to JJ, France muscovy ducks (FF) is another muscovy duck population that is introduced into Hainan province since 1990’s. However, FF grows faster than JJ and is fully covered by white plumage. Mallard (YD), widely accepted as the ancestor of domestic duck [11], is used to confirm whether JJ and FF are also derived from YD. Beijing Duck (BD) has been subjected to intensive selection to provide the raw material of Beijing roast duck dishes. Hong duck (HD) is the hybrid offspring of Beijing duck and sheldrake with higher quality of meat relative to Beijing duck and faster growth rate compared to sheldrake.” in the last paragraph of introduction. Statements like the above always lead to vagueness and deems only four duck populations. Therefore, we added the sentence that “The first one was JJ, which has been introduced as above-mentioned.” before the sentence of “Similar to JJ, French muscovy duck (FF) is another muscovy duck population that has been introduced into Hainan Province since 1990’s” in the revised version (LINE 73). 

Results and discussion：Only basic and necessary results were showed.

What puzzles me is this, you list your results and say they are meaningful, but I can not find any deep discussion. After your sequencing and analysis, some specific SNPs and Indel, GO and KEGG project can illustrate what? I think readers are look forward to in-depth discussions and the specific and functional point.

A: The reviewer provides a constructive suggestion. According to the reviewer, we reread the paper again and added the followed section.

1. The 3.1 section of results and discussion, we changed the statements in the revised version like as follow, “These results indicated that the majority of the identified mutations were located in non-coding sequences (intergenic regions or intronic regions), and only a small proportion of them fell in exon sequence, showing that only very few variants along with duck genome could exert functional effects on protein function. Our results were consistent with those of Xu et al. [8] in ducks and Carneiro et al. in rabbits [6].” (LINE 222-227).

2. The 3.3 section of results and discussion, we inserted that “Interestingly, the genetic relationship between JJ and YD was closer than that between YD and HD or BD. Since YD (mallard) was considered as the ancestor of HD and BD, we hypothesized that JJ and FF were differently originated from HD and BD.”(LINE 272-275).

Tables：All your tables do not meet the standards (Three-line table) , required notes missing. In Table 3, what is the meaning of different background notes?

A: We changed all tables to Three-line tables and added the required notes in the revised manuscript. The added notes are “BD1- BD3, Beijing duck, individual 1-individual 3; FF1- FF3, French muscovy duck, individual 1- individual 3; HD1- HD3, Hong duck, individual 1-individual 3; JJ1- JJ3, Jiaji duck, individual 1-individual 3; YD1- YD3, mallard, individual 1- individual 3.” for Table 1, “SNPs mean single nucleotide polymorphisms; Indels mean short insertions and deletions.” for Table 2, and “Grey represents the comparison of JJ vs. BD and the secretion related pathways in the significant enriched KEGG pathways. Light green means the comparison of JJ vs. BD and the secretion related pathways in the significant enriched KEGG pathways.” for Table 3.

Reference ：Format is also nonstandard. Journal Title, abbreviation, PubMed Central PMCID and so on.

 A: A: We have changed the style of our manuscript to meet PLOS ONE's style requirements according to http://www.journals.plos.org/plosone/s/file?id=wjVg/PLOSOne_formatting_sample_main_body.pdf, including the format of reference.

---

## [Decision Letter · Decision Letter 1]

28 Jan 2020

Genetic characteristics of Jiaji duck by whole genome re-sequencing

PONE-D-19-18411R1

Dear Dr. Xu,

We are pleased to inform you that your manuscript has been judged scientifically suitable for publication and will be formally accepted for publication once it complies with all outstanding technical requirements.

With kind regards,

Marc Robinson-Rechavi

Academic Editor

PLOS ONE

Additional Editor Comments (optional):

Reviewers' comments:

Reviewer's Responses to Questions

**Comments to the Author**

1. If the authors have adequately addressed your comments raised in a previous round of review and you feel that this manuscript is now acceptable for publication, you may indicate that here to bypass the “Comments to the Author” section, enter your conflict of interest statement in the “Confidential to Editor” section, and submit your "Accept" recommendation.

Reviewer #2: All comments have been addressed

Reviewer #3: All comments have been addressed

2. Is the manuscript technically sound, and do the data support the conclusions?

Reviewer #2: Yes

Reviewer #3: Yes

3. Has the statistical analysis been performed appropriately and rigorously? 

Reviewer #2: Yes

Reviewer #3: Yes

4. Have the authors made all data underlying the findings in their manuscript fully available?

Reviewer #2: Yes

Reviewer #3: Yes

5. Is the manuscript presented in an intelligible fashion and written in standard English?

Reviewer #2: Yes

Reviewer #3: Yes

6. Review Comments to the Author

Reviewer #2: There is no other comments for this paper. I think the manuscript is now ready for publication in PLOS ONE.

Reviewer #3: After checking all responses for me and the revised manuscript, I consider that the current revision is suitable for publication as it.

7. PLOS authors have the option to publish the peer review history of their article (what does this mean?). If published, this will include your full peer review and any attached files.

Reviewer #2: No

Reviewer #3: No

---

## [Editor Report · Acceptance letter]

30 Jan 2020

PONE-D-19-18411R1 

Genetic characteristics of Jiaji Duck by whole genome re-sequencing 

Dear Dr. Xu:

I am pleased to inform you that your manuscript has been deemed suitable for publication in PLOS ONE. Congratulations! Your manuscript is now with our production department. 

With kind regards,

on behalf of

Prof. Marc Robinson-Rechavi 

Academic Editor

PLOS ONE